# FROM GAMEPLAY TO SYMBOLIC REASONING

**Fei Wang, Tiark Rompf**
Department of Computer Science
Purdue University
West Lafayette, IN 47906, USA
`{wang603,tiark}@purdue.edu`

## ABSTRACT

Despite the recent successes of deep neural networks in various fields such as image and speech recognition, natural language processing, and reinforcement learning, we still face big challenges bringing the power of numeric optimization to symbolic reasoning. Researchers have proposed different avenues such as neural machine translation for proof synthesis, vectorization of symbols and expressions for representing symbolic patterns, and coupling of neural back-ends for dimensionality reduction with symbolic front-ends for decision making. However, these initial explorations are still only point solutions, and bear other shortcomings such as lack of correctness guarantees. In this paper, we present our approach of casting symbolic reasoning as games, and directly harnessing the power of deep reinforcement learning in the style of Alpha(Go) Zero on symbolic problems. Using the Boolean Satisfiability (SAT) problem as showcase, we demonstrate the feasibility of our method, and the advantages of modularity, efficiency, and correctness guarantees.

## 1 INTRODUCTION

Deep neural networks (DNN) have experienced tremendous success in a large range of applications, including image and speech recognition, natural language processing, and gameplay via reinforcement learning. On the other hand, many discrete automated reasoning applications depend on symbolic manipulation, based on conceptual abstraction, causal relationships and logical composition. Despite the fast advances in DNN, researchers are only making baby-steps to extend the power of DNN to symbolic reasoning, in the quest for general AI.

Researchers have proposed different avenues for applying DNN to symbolic reasoning. One simple avenue is to cast symbolic reasoning as a neural machine translation problem, from propositions (as sequentialized tokens) to proofs (Sekiyama et al., 2017). Using a SEQ2SEQ model, Sekiyama et al. were able to generate many correct proofs, but some proof terms were syntactically malformed or semantically incorrect. Another avenue is to learn how to directly manipulate symbols or apply symbolic rewriting rules (Cai et al., 2017a;b). This method depends on encoding symbols, rewriting rules, and parsed symbolic expressions as vectors, which are used to train a deep feed-forward network for learning the symbolic patterns in the vectorized representations. A third avenue is to combine a neural back-end with a symbolic front-end (Garnelo et al., 2016), hoping to bring language-like propositional representations from classical AI to the low level symbol generation from neural networks. In this method, the neural network is only used to generate features from noisy data, which leaves further room to exploit neural networks for decision making.

In this paper, we present another avenue – casting symbolic reasoning problems directly as gameplay – that leverages the full decision making power of neural networks through deep reinforcement learning (Sutton & Barto, 1998). This direction has several appealing properties. First of all, our gameplay approach is modular, so that we can test different neural network architectures with little adaptation, on different symbolic reasoning problems casted as games. Secondly, by following the possible states and rules in a game, the neural networks cannot make invalid moves or generate incorrect solutions. Lastly, our method uses neural networks for central decision making, yet still maintains symbolic explanations of those decisions in the game setting.

We use the Boolean Satisfiability (SAT) problem as our showcase. SAT is the problem of deciding whether a given boolean formula can be satisfied by any variable assignment. SAT was the first problem proven NP-complete, and SAT solvers serve as fundamental reasoning engines in automated theorem proving, program verification, and program synthesis, often extended to SAT modulo theories (SMT). Most SAT solvers build on the Conflict Driven Clause Learning (CDCL) algorithm (Silva et al. (2009), shown in Figure 1), which is a typical symbolic reasoning process that can be casted as a game of controling the branching decisions. We present our results using two deep reinforcement learning algorithms, DeepQ (Mnih et al., 2013) and Alpha(Go) Zero (Silver et al., 2017a;b), and evaluate the quality of our methodology by comparing with a known high-performance branching heuristics, VSIGS (variable state independent decaying sum, Moskewicz et al. (2001)).

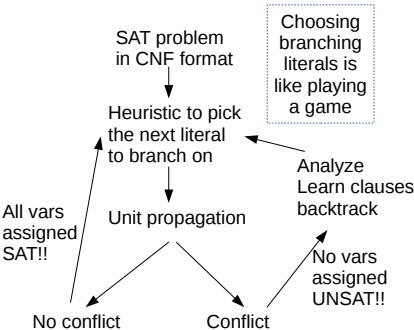

Figure 1: SAT solver algorithm CDCL casted as a game

## 2 DESIGN, EXPERIMENTS, AND RESULTS

We represent SAT problems in Conjuctive Normal Form (CNF) as sparse adjacency matrices, with the rows being the clauses, and the columns being the variables. If a clause contains a variable, the matrix value at the corresponding index is either $(1, 0)$ or $(0, 1)$, depending on the polarity of the variable. The matrices are then fed into a simple convolutional neural network. We use the same CNN architecture for DeepQ as in the OpenAI Baselines[1], but adapted it slightly for Alpha(Go) Zero, to mimic the model in the AlphaGo Zero paper (Silver et al., 2017b), while maintaining the same level of model complexity. We extended the MiniSat[2] SAT solver, building on the basic SAT solving logics based on VSIGS branching heuristics. We engineered the implementation so that it is able to act like a game environment to play with (a gym, in OpenAI terminology). The SAT-game will ask for every branching decision and allow simulations for Monte Carlo Tree Search (MCTS) as needed by the Alpha(Go) Zero algorithm. A simulation returns the would-be state given a sequence of branching decisions without irreversibly changing the game state. In our experiments, we used randomly generated 3-SAT problems (91 clauses and 20 variables, half satisfiable and half unsatisfiable) as benchmarks (see SATLIB[3]).

The DeepQ algorithm tries to learn a $Q$ value (optimal total reward) given a state and action pair $(s, a)$. For a tuple $(s, a, r, s')$ (making action $a$ at state $s$ reaching state $s'$ with reward $r$), it tries to minimize the difference between $Q(s, a)$ and $r + \gamma * \max_{a'} Q(s', a')$, where $\gamma$ is a normalization factor and $a'$ represents any valid action from $s'$. The Alpha(Go) Zero algorithm tries to learn, for each state $s$, the degree of interest (as a probability vector $pi$) of all valid actions, and the state quality ($v$ in $[-1, 1]$). At each state, the algorithm explores interesting actions guided by state quality in MCTS, which provides a stronger estimation of $pi$ and $v$ by counting how many times each action is used in simulation, and in the final result of the game.

We present the experimental results in Figure 2, with the x-axis showing different model checkpoints (model-0 is the random initialization), and the y-axis showing the average number of branching deci-

---

[1]https://github.com/openai/baselines
[2]https://github.com/niklasso/minisat
[3]http://www.cs.ubc.ca/~hoos/SATLIB/benchm.html

sions needed to solve the SAT-game (the lower the better). In our experiment, the DeepQ algorithm converged on the training set (dark blue line), but failed to generalize to the testing set (red line). On the other hand, the Alpha(Go) Zero algorithm converged to near optimal performance on the training set (yellow line), and generalized quite well to the testing set (green line). All converged models outperformed MiniSat heuristics (purple and light blue). It is also worth noting that the training set is only 32 SAT problems, while the testing set is 200 SAT problems.

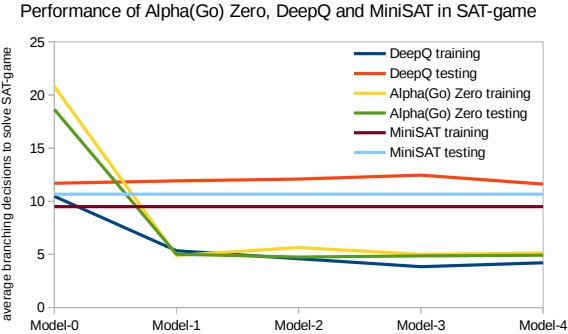

Figure 2: The performance of Alpha(Go) Zero, DeepQ and MiniSat in SAT-game

# 3 DISCUSSION AND FUTURE WORK

In this paper we evaluated our unique approach to casting symbolic reasoning as games, with the goal of harnessing neural networks for decision making through reinforcement learning. Compared with other approaches of combining DNN and symbolic reasoning, our approach is modular, efficient, and guaranteed to be correct. Using SAT problems as our showcase, we compared the DeepQ and Alpha(Go) Zero algorithms with human-engineered heuristics in MiniSat. We observed that both reinforcement learning algorithms can converge on the training set, but only Alpha(Go) Zero generalized to the testing set.

It is interesting to discuss why Alpha(Go) Zero outperformed DeepQ in generalization (which is important for general AI). Our speculation is that DeeqQ learns a $Q$ score that is very specific to a $(s, a)$ pair. On the contrary, Alpha(Go) Zero appears to learn a general degree of interest $pi$ and a relative state quality $v$. While DeepQ tries to condense the knowledge in one value ($Q$) for each state, Alpha(Go) Zero always uses MCTS to explore a tree of local states and leverage knowledge from many states. Arguably, Alpha(Go) Zero already reasons on an abstract and symbolic level, by learning $pi$ as the degree of interest, and $v$ as the state quality, which has been described as "intuition", similar to how humans reason about games.

One important issue of combining DNN with symbolic reasoning is the difficulty of interfacing between a symbolic world where most representations are sets, trees, and graphs, and the numerical world where everything is vectors, matrices, and tensors. Many efforts (including this one) use sparse vectors as a simple translation. However, progress in natural language processing (NLP) might offer some alternatives, such as word embeddings (Mikolov et al., 2013; Pennington et al., 2014) as a non-trivial way of vectorizing words in a condensed and meaningful manner. On the other hand, progress in more advanced neural network architectures, such as TreeLSTM (Tai et al., 2015), might facilitate the interfacing as well.

Needless to say, our SAT benchmarks are too small to be practically useful. However, we argue that our showcase is a proof-of-concept for harnessing numerical optimization for symbolic problems. Potential future work includes exploring methods to scale up to larger SAT problems, and addressing other symbolic reasoning problems in this style of gameplay.

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
