# OpenReview forum: "From Gameplay to Symbolic Reasoning"
_ICLR.cc/2018/Workshop — Reject_

### Official Review · AnonReviewer3 · 2018-02-25
**Writing the rule-book of real-life "games" is the hard part!**

**Rating:** 6
**Confidence:** 3

**Review:**

This extended abstract proposes to approach symbolic problems with neural networks by formalizing problems as a games, and then using deep RL methods to teach a network to play the game. The idea is illustrated through the Boolean Satisfiability problem, where an AlphaGo-like system is shown to outperform deep Q learning and a heuristic method in terms of generalization to the test set.

The proposal is interesting, but I find it problematic for the following reason. To formulate a problem as a game, you must be able to formalize it well enough that you can exhasutively define the set of rules. However, the history of GOFAI suggests that, in most interesting real-life domains, experts are unable to craft such rules by hand, which is the main reason why the field eventually shifted to learning-based systems, such as neural networks. How would your approach scale up to, say, handling natural language--arguably a symbolic domain, but one where, despite centuries of trying, no linguist has ever come up with an exhaustive "rulebook". I would really like the authors to discuss this point.

I assume most ICLR attenders will not be familiar with symbolic approaches to SAT. So, it would be good to describe how the problem is formulated in more details. Also, please give us a sense of how good MiniSAT is, with respect to other symbolic solvers: how impressive is it that AlphaGo outperforms it?

Finally, it would be good to have (in an Appendix) more details on the deep RL systems and how they were trained.

---

### Official Review · AnonReviewer1 · 2018-03-10
**Nice ideas, but still preliminary in doing any kind of symbolic reasoning**

**Rating:** 7
**Confidence:** 4

**Review:**

The authors focus on the problem of learning symbolic reasoning from examples by deep neural networks. This sort of problem has been discussed for a way, and has received significant attention lately, as noted by the authors. The paper is thus quite relevant. It is well written and quite clear, packing quite a bit of material in a short space.

The solution presented by the authors is to learn to prove things by understanding the act of proving as a game where one tries to find policies that take one to a proof. In this analogy, to learn to prove is to learn a game, and the authors try recent algorithms focused on game learning to the setting of symbolic reasoning. This is an interesting approach and I think it is a promising path that deserves some attention.

However, the paper has its weaknesses. For one thing, the experiments are interesting but very small; the authors acknowledge this and it is perhaps natural to have small experiments in preliminary work. But the problem is more serious: it seems to me that by learning SAT solving, the authors are just learning heuristics for SAT solvers. Now, it is not clear that heuristic learning means really "symbolic reasoning learning". Perhaps I missed something in this regard, but the text seems clear in its explanation.

Also, given that SAT solving is now highly advanced, getting to learn and test with SAT problems of some hundred variables seems frustratingly little to advance the cause of symbolic reasoning. And on top of that, it seems to me that the kind of reasoning that is learned here is always approximate reasoning (the authors say that reasoning is "guaranteed to be correct", but how can this be guaranteed...?), not exact reasoning. So the correct comparison here is with large approximate SAT solvers; in that case, do the learned policies win? This is something I urge the authors to try.

Overall, I find the work relevant, well written, and with some original contributions. So it is worth publishing, but it is really very preliminary and may not work well in more testing. Significant work is still to be done here.

Finally, there are some problems in the references with capitalization (atari, go).

---

### Official Review · AnonReviewer2 · 2018-03-11
**Interesting approach but too vague/little**

**Rating:** 4
**Confidence:** 3

**Review:**

The paper uses a way to cast SAT solving as a Conflict Driven Clause Learning game, and presents results deep models trained on these games. The input features consist of encoding the SAT problems in Conjunctive Normal Form as sparse adjacency matrices. The training algorithms are either DQN or Alpha(Go)Zero-like. They compare results to a MiniSAT baseline, on a train and test set taken from SATLIB. The results are: DQN overfits and has worse error than baseline on test. AlphaZero has twice better (i.e. less) average branching decisions than baseline on train and test.

The approach is interesting, but the paper is too vague and includes too little results so that one can conclude that the promising positive results are not due to MCTS, or to the baseline being weak. A short description of the model(s) and a description of the differences between train and test time (if any) would help.

Conjuctive -> Conjunctive

---

### Decision · Program_Chairs · 2018-03-20
**ICLR 2018 Workshop Acceptance Decision**

**Decision:**

Reject

**Comment:**

Based on the reviews, this paper has not been accepted for presentation at the ICLR workshop. The results were too small scale and quite preliminary, even for the workshop. However, the conversation and updates can continue to appear here on OpenReview, and we look further for improvements!